# Self-Perceived Health among Migrants Seen in Médecins du Monde Free Clinics in Europe: Impact of Length of Stay and Wealth of Country of Origin on Migrants’ Health

**DOI:** 10.3390/ijerph16244878

**Published:** 2019-12-04

**Authors:** Simon Jean-Baptiste Combes, Nathalie Simonnot, Fabienne Azzedine, Abdessamad Aznague, Pierre Chauvin

**Affiliations:** 1Univ Rennes, EHESP, CNRS, ARENES–UMR 6051, 35000 Rennes, France; fabienneazz@hotmail.com (F.A.); abdessamad.aznague@gmail.com (A.A.); 2French Collaborative Institute on Migration, 93322 Aubervilliers, France; 3Médecins du Monde–Doctors of the World, International Network, 75018 Paris, France; nathalie.simonnot@medecinsdumonde.org; 4Department of Social Epidemiology, Sorbonne Université, INSERM, Institut Pierre Louis d’Epidémiologie et de Santé Publique (IPLESP), 75012 Paris, France; pierre.chauvin@inserm.fr

**Keywords:** migrant health, length of stay, Médecins du Monde, self-perceived health, migration, Human Development Index

## Abstract

Health of migrants is a widely studied topic. It has been argued that migrant health may deteriorate over time. Though migrants are a “hard to reach” population in survey data, this paper builds on a unique dataset provided by Médecins du Monde from five countries. We study self-perceived health (SPH) in connection with socio-economic and demographic factors and length of stay. Results differ for men and women. Compared to other documented migrants, asylum seekers have a 50–70% greater chance of having worse health. Migrants with better living conditions have a 57–78% chance of being in better health. Male migrants with a job have between a 82–116% chance of being in good health. The probability for women from poorer countries to have a better physical SPH after three months of residing in the host country is six-fold that of women from richer countries. This paper contributes widely to the knowledge of health of migrants. Contrary to other evidence, health of women migrants from poorer countries tends to improve with length of stay.

## 1. Introduction

The health of migrant populations is an emerging trend in public health research which has produced mixed results [1,2,3]. The Healthy Migrant Effect (HME)—which describes an empirically observed lower morbidity and/or mortality of migrants from certain countries of origin, relative to the majority population in the host countries (usually in the industrialized world)—remains a debatable question. Inconsistent evidence shows that migrants can be either in better or worse health than the population of their host country [4,5,6,7,8,9,10,11]. Many factors can contribute to these inconsistent results, such as the host country itself (migrants in North America and southern Europe are in better health than migrants in northern Europe [12]), migrants’ social integration in the host country (migrants living in host countries where they are more easily integrated tend to be in better health [7]), or their length of residence (migrants’ health worsens over time [4,13]). On average, migrants tend to live in worse conditions than their host populations [14]. In Europe, the health of migrants who have settled during recent years may be worse than those of migrants who arrived during the 1970s when labor migration was more common, the labor market was more in demand, and borders were more open [14].

Many researchers have explained the better health status of recent migrants in terms of a self-selection process: migration circumstances result in only the healthiest candidates having privileged access to emigration. Compared to the population of industrialized countries, migrants from developing countries may also have healthier lifestyle habits (eating behaviors, smoking, physical activity, etc.). However, after a certain period of time in the host country, cultural integration, difficult employment and living conditions, lower social status, and the weakening of social and familial links may have a negative impact on their health [15,16]. Access to healthcare is also more of a challenge for migrants than for nationals for many reasons: lower healthcare literacy, reduced ability to assert their rights, and discrimination in healthcare services [17]. More recently, in light of the differences in migrants’ health according to their country of origin, some authors have argued that migration from a society in an earlier phase of the health transition to a society in a more advanced phase has a positive effect on migrants’ health [3]. A recent article also showed that the convergence of health of migrants differs by wealth of the country of origin [13], and another article showed that health differences are partly explained by the relative educational attainment in the country of origin [18].

In this paper, we add to the literature on healthy migrant effect by testing whether the effect of length of stay on migrant’s health differs by country of origin. We compare the health of migrants who sought care and see if there is a correlation between length of stay and health. We hypothesize that the effect of length of stay on the health of migrants differ by the wealth of the country of origin. Following the literature on the healthy migrant effects, we expect health to worsen with time but at different rates for migrants coming from a poorer or wealthier country [13].

To assess health, we used self-perceived health (SPH). There is a strong body of evidence showing that SPH predicts mortality and morbidity as well as a medical diagnosis [19,20,21,22,23,24,25]. SPH is a worse predictor of mortality among higher socio economic statuses [26] which might be due to richer and more educated people reporting worse health when given a health state to assess [27]. SPH has been shown to be sensitive to ethnicity [28], but Chandola and Jenkins show that there is no joint effect of SPH and ethnicity predicting morbidity [29], thus showing that the predictability of SPH is not modified by ethnicity. Moreover, SPH has been widely used to compare health in different countries [30,31]. A recent article shows that the effect of being a non-EU citizen in the EU on SPH is similar on chronic conditions or reporting limitations in daily life [7].

We test whether length of stay and the wealth of the country of origin have an impact on migrants’ health. In order to do so, we analyzed data collected by Doctors of the World/Médecins du Monde from free clinics in five countries to estimate the respective effects of the wealth of the country of origin and the length of residence in the host country on the perceived health of adult migrants (separately for men and women). We took into account various living conditions in the host country. Migrants from wealthier countries were expected to be in better health when they first arrived compared to migrants from poorer countries. After a while in the country of residence, we expected living conditions to level off migrants’ health, i.e., the length of stay in the host country would modify the effect of country of origin on migrants’ health.

## 2. Materials and Methods

### 2.1. Studied Population

The main mission of Doctors of the World/Médecins du Monde (MdM) is to provide access to healthcare through free social and medical services for people who face barriers to the mainstream healthcare system. MdM works mainly with people confronted with multiple vulnerabilities affecting their access to healthcare, including homeless people, drug users, and destitute nationals, as well as migrant EU citizens, sex workers, undocumented migrants, asylum seekers, and Roma communities. MdM programs aim at empowerment through the active participation of user groups as a way of identifying health-related solutions and combating the stigmatisation and exclusion of these groups.

The programs which collected our data are free clinics run by MdM or its partners, which offer primary healthcare consultations as well as social support and information about the healthcare system and patients’ rights with regard to accessing healthcare. Ultimately, these programs aim to help patients reintegrate into the mainstream healthcare system, where this is legally possible. The patients seen by MdM are therefore those individuals who have found health care via MdM, which means that they were in need of care and ineligible (or so they thought) for medical care in the common law healthcare system, either because they cannot afford them or for other administrative reasons.

MdM programs are run predominately by volunteers (90% are volunteers and the remaining 10% are paid staff, of which around 90% are health professionals—including nurses, general practitioners, midwives, dentists, medical specialists, and psychologists—and 10% are social workers, support workers, mediators, and translators). The MdM International Network has developed a quantitative and qualitative information system that includes systematic patient data collection and annual statistical analysis, narrative patient testimonies, de jure and de facto legal analysis of healthcare systems, as well as identification of best practices.

For this analysis, we based our sample on the total population seen by MdM. In the end, we included 1356 adult migrants who consulted an MdM volunteer at one of the free clinics run by MdM or its partners in five countries for the first time in 2014. These free clinics are located in Munich (Germany), Alicante, Bilbao, Malaga, Seville, Tenerife, Valencia, Zaragoza (Spain), La Chaux de Fonds, Neuchâtel (Switzerland), Istanbul (Turkey), and London (United Kingdom). Unfortunately, we could not include patients seen in Greece as their asylum status and length of stay in Greece were not available.

### 2.2. Data Collection

MdM volunteers (doctors, nurses, social workers) collected data through a medical and social questionnaire administered to every new adult patient. The medical questionnaire collected information on perceived health, vaccination, pregnancy and contraception, experiences of violence, history of Hepatitis B, Haemoglobin C and Human Immunodeficiency Virus testing, reasons for consultation, and diagnoses at the end of the consultation. The social questionnaire collected information on housing conditions, occupation and resources, administrative situation, health coverage, and obstacles to accessing healthcare. No ethics committee approval was required for this study as it uses socio-medical data routinely collected by health professionals. The medical records used in the study were collected and archived under the supervision of health professionals and according to the national laws and regulations of each country. The data used by the authors of this study was fully anonymised.

### 2.3. Outcomes

We used two indicators of self-perceived health (SPH) in order to distinguish between physical and mental health. We used the answers to the questions “How is your physical health state?” and “How is your emotional and psychological health state?”, respectively, and categorized both of them (very good and good versus fair, bad, or very bad; the latter three categorizations are referred to as “deteriorated self-perceived health” in the rest of the paper).

### 2.4. Covariates

Two characteristics relating to patients’ migration status were used in the analysis: residence status (residence permit, undocumented, or asylum seeker) and length of stay in the host country (in three categories: <3 months, between 3 months and 5 years, more than 5 years). We also used four indicators of socioeconomic status (SES): housing conditions (unstable/stable), income satisfaction (not enough/enough for basic needs), job (yes/no), and health coverage (yes/no). In the second part of the analysis, we used the Human Development Index (HDI) of patients’ countries of origin [32]. The cut-off points for length of stay and HDI were studied in a sensitivity analysis that can be found in the annexes.

### 2.5. Statistical Analysis

Since self-perceived health is known to differ between men and women [33,34]—although it should be noted that very few studies have analysed migrant health in particular by sex [35]—all of our data has been analysed by sex. Comparisons of men and women’s descriptive statistics are performed using a proportion test.

Firstly, we estimated a logistic regression model which included demographics (age in quartiles, country of residence, i.e., where people were interviewed and region of origin), migration status (residence status and length of stay in the host country) and SES (housing conditions, income satisfaction, job, and health coverage).

Secondly, we created a binary variable indicating whether or not the person is from a wealthy country (HDI above 0.5), using the cut-off point of 0.5 following Chaix et al. [32]. We tested whether the HDI of the country of origin is linked to our outcomes in men and women (see full results online), then tested whether the effect of length of stay in the host country modified these associations. We defined two categories of length of stay according to the results of the first model (<3 months, ≥3 months). In this final model, the Human Development Index of the country of origin was added as a modifier to test the hypothesis that individuals coming from a more developed country have better SPH [36] and that the change in SPH is influenced by the amount of time spent in the host country.

## 3. Results

### 3.1. Descriptive Statistics

The final data shows the results from 577 women and 767 men for physical SPH, and 576 women and 765 men for mental SPH.

Table 1 shows the descriptive statistics of each explanatory variable cross-referenced with each dependent variable for men and women. We have presented the proportion of people who are in good or very good health. The results show that women in the first age quartile are more likely to be in better mental health than men (31.9% vs. 20.8%), and that men residing in Spain are more likely to be in better physical health than women (50.0% vs. 33.6%). Men from Africa are more likely to be in better physical health than women (37.0% vs. 26.6%). With regard to undocumented migrants, women are more likely to be in better mental health (25.1% vs. 15%) while men are more likely to be in better physical health (43. 3% vs. 33.0%). When living in unstable accommodations, men are more likely to be in better physical health than women (32.5% vs. 22.7%). When income is deemed insufficient, men are more likely to be in better health than women (33.6% vs. 26.9%). When migrants have a job, women are more likely to be in better mental health than men (30.0% vs. 20.2%). Finally, when migrants have health coverage, men are more likely to be in better health than women (39.0% vs. 28.6%).

### 3.2. Multivariate Models

We will now discuss multivariate models in order to explain good health (versus deteriorate health as a reference), using logistic regression models.

#### 3.2.1. Asylum Seekers

One significant result of our study is the finding that asylum seekers are in worse health than other documented patients. The results for both models and both sexes are negative but are significant only for female asylum seekers’ physical SPH (0.50 [0.27; 0.94]) and male asylum seekers’ mental SPH (0.32 [0.17; 0.60]). The physical SPH of female asylum seekers is twice as poor as that of other documented women. Male asylum seekers are three times more likely to have worse mental SPH than other documented men.

#### 3.2.2. Living Conditions

Living in stable accommodation is associated with better health. While this variable is positive for both models and both sexes, it is significant for women’s physical and mental SPH (1.78 [1.18; 2.70]; 1.96* [1.26; 3.08], resp.), and men’s mental SPH (1.57 [1.04; 2.39]). Women living in stable accommodation are twice as likely to be in better mental SPH than women who do not have stable accommodation.

Finally, work is also associated with better health. This is significant for both the physical and mental SPH of men. Men in employment are twice as likely to be in better mental SPH than men who are not in employment.

#### 3.2.3. Length of Stay (LOS)

In Table 2 we present the results for the following thresholds of length of stay: less than three months, between three months and five years, and more than five years. We tried different sensitivity analyses following different research articles [37,38]. We tested the following durations of residence: less than three months, between three months and five years, between five years and 10 years, and more than 10 years (All results are available online Appendix A.). The only significant result is the finding that female migrants who have spent more than three months in their host country have better physical SPH than the women who have stayed for less than three months. The effect of length of stay on mental and physical SPH of men and mental SPH of women were not significant.

One interesting hypothesis is that women’s physical SPH seemed to decrease in women who have stayed in their host country for more than 10 years. In a sensitivity analysis (Appendix A) document we tested different specifications to test this hypothesis of an inverted U shape effect of LOS on health. The resulting assumption is that health improves as the host country is wealthier than country of origin, then deteriorates as migrants tend to live in poor conditions and suffer discrimination. However, although this result appears in some specifications, it is not consistent across models.

#### 3.2.4. Wealth of Country of Origin and Length of Stay, Effect Modifier

Next, we tested the effect of the Human Development Index, firstly to determine its impact on health and secondly to test whether the wealth of the country of origin is an effect modifier for Length of Stay (LOS). Table 3 shows the results for women’s physical SPH only. Full results are available online. Introducing effect modifiers in male models did not change the general finding that length of stay has no effect for males (See Appendix A). Interaction models with LOS in three categories do not provide much more information than LOS in two categories (less than three months and more than three months), therefore we have only presented interactions for the latter. The introduction of the effect of wealth (effect modifier) does not change the results for the other covariates shown in Table 2.

The effect of the wealth of the country of origin shows that women from wealthier countries have better physical SPH when they have stayed for less than three months compared to women from poorer countries. The physical SPH of women from wealthier countries does not change with time spent in the country. Women from poorer countries see their physical SPH improve with a length of stay longer than three months. Similarly, to LOS, we tried different specifications of the wealth cut-off point and while some results were significant for men, they were not robust when tested with different cut-off point specifications. However, the effect for women was similar regardless of the wealth cut-off point (0.5, 0.6, or 0.7) (Appendix A.).

## 4. Discussion

This study of migrant patients visiting MdM health centres in Europe provides an interesting insight into the health status of vulnerable migrants. This population is not homogenous and different factors impact their health.

Firstly, we have learned that the effect of being an asylum seeker compared to other documented migrants has a negative effect on women’s physical SPH and men’s mental SPH.

Secondly, when patients have better living conditions they tend to be in better health. Employment has a positive effect on both the physical and mental SPH of men. Stable accommodation has a positive effect on women’s SPH (physical and mental) and men’s mental SPH.

Lastly, to answer our research question, the duration of stay in host countries has contradictory effects depending on the wealth of the country of origin. Women from poorer countries, as measured by the Human Development Index, see their physical SPH improve after three months of residing in the host country, whereas there is no effect for those from wealthy countries.

### 4.1. Perceived Health

We used indicators of self-perceived health (SPH) to assess migrants’ health. There is a strong body of evidence showing that SPH predicts mortality and morbidity as well as medical diagnosis [19,24]. It is a better predictor for underprivileged people [26] and/or those with lower levels of education. SPH has been shown to vary according to ethnicity [28]; however, Chandola and Jenkins have shown that there is no joint effect of SPH and ethnicity predicting morbidity [29], i.e., the predictability of SPH is not modified by ethnicity. A recent article showed that the effect of being a non-EU citizen living in the EU on SPH was similar to that of people who suffer from a chronic condition or report limitations in daily life [7].

### 4.2. The Health of Asylum Seekers

The literature that concerns the health status of asylum seekers, refugees, and undocumented migrants shows that these three administrative categories face risk factors for mental health disorders during premigration (persecution, armed conflicts, and economic hardship), perimigration (different kinds of violence, life-threatening conditions, separation from family and support networks), and post migration (for example, see [17,39,40]). Once in the host country, while refugees struggle to fully integrate into society, asylum seekers also experience feelings of uncertainty about their asylum applications (the longer the procedure, the worse their mental health) and fear of detention. Toar et al. [39] have shown that asylum seekers have a higher level of self-reported post-traumatic stress disorder and depression/anxiety compared to refugees.

Our results show the negative effect of being an asylum seeker on women’s physical health and men’s mental health compared to other documented patients. These results are consistent with literature showing the significant impact of the stress of asylum procedures and living through a period of uncertainty [40].

Administrative situations can change over time. For example, asylum seekers may previously have been undocumented, refugees may have been asylum seekers, and migrants may become refugees or legal residents through other procedures. As such, in order to understand how an administrative situation affects health, qualitative research may be more appropriate for understanding the process by which health deteriorates. A combination of qualitative and quantitative methods can help to provide insight into the process that results in individual health status; for France, see Cognet et al. [15].

### 4.3. Jobs: Labor Market and Socio-Economic Positions

The role of social class in health inequality has been well-documented [41]. On the topic of migrant populations, Gosselin et al. [42] have shown that sociodemographic factors in the country of origin have little impact on the settlement of sub-Saharan African migrants in France. However, Borrell et al. [43] have studied how social class in host countries may mediate the impact of migration status on health. Moreover, a recent paper by Ichou and Wallace linked data from country of origin with host country and were therefore able to take into account a selection effect of those with higher education into migration. They showed that healthy migrant effect wears off by at least 30% for women when taking into account relative educational achievement in the host country [18].

Most research on health and being economically active focuses on the detrimental effect of working conditions on health [44,45,46]. Others have shown that there is a selection effect of those in good health into work [47], and at a macro-economic level that where there is better health there is more widespread labour supply [48]. There is also evidence that more permanent employment positions have a beneficial impact on individuals’ mental health [49]. Labor force participation reduces the risk of mortality, and being active lowers the risk of mortality when unmarried [50]. It is a public health issue to facilitate access to the labor market in order to improve the health of deprived communities [51,52,53]. Employment conditions for migrants and refugees are usually more strenuous than for host populations [54,55]. However, this may depend on the host country. In Australia, a country that selects migrants based on a point based system, native-born people are more likely to work in agriculture and construction, jobs that are more strenuous [56]; this is not the case in Spain and Italy [57]. Studies on migrant workers have shown that migrants and refugees are at higher risk of occupational exposure, injury, and illness [55]. This is a result of the relegation of migrants to the most dangerous jobs and the most dangerous tasks within these jobs. It is also a consequence of the lack of safety training and unstable jobs, and fear of reprisal for demanding better conditions or reporting an injury or illness. Finally, it may also be the consequence of linguistic and cultural complexities that minimize the effectiveness of training [58]. As such, migrants in employment are in better health than those who are not, though those in employment suffer from detrimental working conditions.

In correspondence with these studies, our results show the positive impact of employment on men’s SPH (both physical and mental). There is no significant evidence for the same impact on women’s SPH. The population we have studied is specific as it cannot benefit from social protection [59]. Furthermore, according to the 2015 MdM report [60] made with the same data, 66% of the patients do not have the right to reside which means that they also do not have the right to work. As the remaining 34% of the people who have the right to reside do not necessarily have the right to work, 66% is therefore an underestimation of those who do not have the right to work. Employment is the primary way to receive income in Western societies and can be perceived positively, even if the salary is low (91.3% of MdM patients were living below the poverty line, and this figure does not take into account the number of people living on this income). Therefore, we argue that access to the job market should be eased for migrants.

### 4.4. Housing: Living Conditions

Fazel et al. [61] have shown that homeless individuals have worse mortality outcomes than the general population in Europe and the USA. In these countries, various programs have been developed to provide stable and safe accommodation for the homeless. Program evaluations show the positive impact of housing on health, but few research studies [62] have been developed to understand the impact of unstable accommodation on physical and mental health. Robert and Vanoni [62] argue that poor housing increases the risk of health issues and violence. A study of female patients in Médecins du Monde in the Paris region show that 44% had experienced violence in their host country and 6% had been raped. Among female victims of violence, 55% reported negative effects on their health. Violence may also be enacted by close acquaintances such as the person(s) hosting the migrants [63]. In France, among homeless families who had accommodation, 3.1% left their previous accommodation due to violence or exploitation by the host [64]. Our study shows the positive impact of stable accommodation on both women’s physical and mental SPH and men’s mental SPH.

As argued in the Ottawa Charter, shelter is a fundamental condition for health [65]. Our evidence shows that we can go further than this to argue that shelter must be stable and safe in order to be a fundamental condition for health.

### 4.5. Health: Length of Stay Versus Wealth of Country of Origin

Research has shown that there is a convergence of health states of migrants to the natives’ that differ by wealth of the country [13]. Migrants coming from a country in an earlier phase of the health transition to a society in a more advanced phase experience a positive effect on their health [3].

Our results show that length of stay, and length of stay stratified by wealth of the country of origin, have no impact on men’s health, and the same is true of women’s mental health. Length of stay has an impact on women’s physical SPH (female migrants that have spent between three months to ten years in their host country are 2.5 times more likely to be in better physical health than women who have stayed for less than three months). Women from wealthier countries are six times more likely to be in better health than those from poorer countries on arrival (less than three months). Women from poorer countries are over four times more likely to be in good health when they stay for longer than three months. There is no effect for migrants staying for longer than ten years.

After a little while in their host countries, women from poorer countries see an improvement in their physical health. Our results reveal an interesting sex difference, which may be explained by the more frequent use of health facilities by women.

We tested different models in order to gain information about the impact of length of stay in the host country on migrants’ health. We have shown that LOS has a positive effect on women’s physical SPH. However, the hypothesis of an inverted U shape effect of LOS on health could become a potential object of further study.

### 4.6. Limitations

It should be noted that our study is not representative of the health status of vulnerable migrants. Firstly, our study is limited to MdM patients. It is likely that vulnerable migrants have different networks regarding health, depending on factors such as host country and country of origin. Moreover, some may not have access to health facilities. Secondly, the profile of patients also depends on the health centre. For example, as the health centre in Turkey is managed by individuals from sub-Saharan Africa, it tends to work with migrants from this region predominantly.

As a consequence, the parameters that are obtained are likely to be biased by a self-selection process.

We also only investigated self-perceived health, which is a self-reported health. As argued in the introduction, it is as good as a clinical or objective measure in predicting mortality [19,20,21,22,23,24,25]. However, clinical measures should also be investigated and recorded.

The term “migrants” in the results shown refers to migrants seen by MdM and partners.

We are aware that our findings are not a result of multiple hypothesis testing. We limited our ambitions to one effect modifier only as we are cautious not to test for every possible interaction. Moreover, when testing different sensitivity analyses, we focused on the results that remained robust when tested with different specifications.

### 4.7. Future Studies

Longitudinal cohort studies on migrant populations are required to better understand the impact of work and poor housing on health, taking into account the different types of work and housing and the evolution of these social determinants over their lifetimes. These kinds of studies will also take into account the trajectories of migration, which appears to be a better way to understand migrant health. In those studies, subjective and clinical/objective measures of health should be used to have a better understanding of how the health of migrants may improve or deteriorate. Qualitative studies may also provide insight into migrant pathways. Further studies are required to better understand the factors which affect health in the country of origin as well as in the host country, and which affect migrants at the various stages of their lives.

## 5. Policy Implications

Stable and safe accommodation is a fundamental condition for health. Therefore, we recommend that countries who ratified the International Covenant on Civil and Political Rights, as well as the European Convention on Human Rights, to implement the right to housing.

In light of the positive effect of employment on social integration and its short-term health benefits, we would recommend facilitating access to the job market for migrants.

## 6. Conclusions

Our hypothesis that length of stay has a different impact on health depending on wealth of the country of origin is confirmed. The research shows that women from poorer countries have their health improve after three months in a host country. In addition, a very consistent result is the effect of housing and job markets: when migrants have accommodation or a job, they are in better health.

## Figures and Tables

**Table 1 ijerph-16-04878-t001:** Proportion of people in very good or good health by sex and covariates.

		Good Self-Perceived Physical Health			Good Self-Perceived Mental Health		
	n	Women %	Men %	*p*	Women %	Men %	*p*
Age group							
1st quartile [18;28]	310	30.1	41.1	0.06	31.9	20.8 *	0.05
2nd quartile [28;34]	383	32.5	37.2	0.38	15.3	17.3	0.68
3rd quartile [34;44]	379	24.2	28.5	0.41	27.9	21.0	0.15
4th quartile [44;85]	284	24.1	23.7	>0.99	29.7	26.6	0.64
Total	1356	27.6	33.4		25.7	20.9	
Surveyed Countries							
Germany	46	24.0	28.6	0.98	36.0	19.1	0.35
Switzerland	30	50.0	25.0	0.87	66.7	29.2	0.28
Spain	210	33.6	50.0 *	0.02	36.4	40.0	0.69
Turkey	552	31.0	39.7	0.06	8.0	6.6	0.66
United Kingdom	518	22.2	19.6	0.52	32.1	32.7	0.96
Total	1356	27.6	33.4		25.7	20.9	
Origin							
Middle East	78	11.5	17.3	0.74	11.5	19.2	0.59
Africa	708	26.6	37.0 **	<0.01	13.9	11.6	0.45
Americas	110	40.5	36.1	0.81	33.8	27.8	0.68
Asia	327	25.3	24.9	>0.99	39.0	38.1	0.96
Europe	49	25.9	40.9	0.42	40.7	31.8	0.65
Maghreb	84	30.0	45.5	0.79	40.0	34.1	0.86
Total	1356	27.6	33.4		25.7	20.9	
Residence status							
Other documented	220	30.8	41.4	0.14	32.7	31.9	0.98
Undocumented	548	33.0	43.3 *	0.02	25.1	15.0 **	<0.01
Asylum Seeker	588	21.3	21.2	>0.99	23.3	22.7	>0.99
Total	1356	27.6	33.4		25.7	20.9	
Length of stay							
<3 months	218	19.2	40.3 **	<0.01	24.2	19.3	0.45
3 months–5 years	791	30.3	34.7	0.21	22.3	18.3	0.22
>5 years	346	27.7	25.0	0.66	32.9	29.0	0.50
Total	1356	27.6	33.4		25.7	20.9	
Housing conditions							
Unstable	715	22.7	32.5 **	<0.01	17.1	15.5	0.64
Stable	641	31.8	34.6	0.50	33.1	28.2	0.22
Total	1356	27.6	33.4		25.7	20.9	
Income satisfaction							
Not enough	1227	26.9	33.6 *	0.01	23.0	20.5	0.34
Enough	129	31.8	29.6	0.95	41.2	27.3	0.17
Total	1356	27.6	33.4		25.7	20.9	
Job							
Yes	874	25.3	27.9	0.41	23.8	21.3	0.45
No	482	32.8	42.1	0.05	30.0	20.2 *	0.02
Total	1356	27.6	33.4		25.7	20.9	
Health coverage							
Yes	364	28.6	39.0 *	0.04	33.0	33.0	>0.99
No	992	27.1	31.7	0.14	22.4	17.2	0.06
Total	1356	27.6	33.4		25.7	20.9	

Comparing female proportion to male proportion, *p* values below 1%, and 5% are shown with two, or one stars, respectively.

**Table 2 ijerph-16-04878-t002:** Multivariate analysis of characteristics associated with good or very good perceived physical and mental health, by sex.

		Women	Men	Women		Men	
				Probability of Being in Good Self-Perceived Health			
				Physical	Mental	Physical	Mental
	n	%	%	aOR [95%CI]	aOR [95%CI]	aOR [95%CI]	aOR [95%CI]
Age group							
1st quartile [18;28]	310	36.5	63.5	Ref.	Ref.	Ref.	Ref.
2nd quartile [28;34]	383	41.0	59.0	1.07 [0.61; 1.90]	0.52 [0.26; 1.00]	0.92 [0.60; 1.40]	0.96 [0.55; 1.69]
3rd quartile [34;44]	379	43.5	56.5	0.65 [0.36; 1.17]	0.63 [0.34; 1.15]	0.64 [0.41; 1.00] *	1.08 [0.61; 1.92]
4th quartile [44;85]	284	51.1	48.9	0.66 [0.35; 1.23]	0.59 [0.31; 1.10]	0.65 [0.38; 1.13]	0.89 [0.48; 1.64]
Surveyed Countries							
Germany	46	54.3	45.7	Ref.	Ref.	Ref.	Ref.
Switzerland	30	20.0	80.0	6.27 [0.81; 51.34]	4.84 [0.67; 44.95]	1.12 [0.27; 4.71]	3.10 [0.69; 15.55]
Spain	210	52.4	47.6	0.53 [0.14; 2.12]	0.74 [0.21; 2.62]	2.14 [0.75; 6.76]	3.25 [1.00; 12.91]
Turkey	552	33.9	66.1	0.91 [0.19; 4.45]	0.08 [0.02; 0.38] *	1.54 [0.40; 6.24]	0.11 [0.02; 0.55] *
United Kingdom	518	48.6	51.4	0.38 [0.09; 1.68]	0.30 [0.08; 1.20]	0.61 [0.16; 2.43]	0.97 [0.23; 4.65]
Region of Origin							
Middle East	78	33.3	66.7	Ref.	Ref.	Ref.	Ref.
Africa	708	37.7	62.3	2.87 [0.85; 13.37]	2.20 [0.59; 11.35]	1.20 [0.53; 2.92]	0.82 [0.35; 2.06]
America	110	67.3	32.7	9.15 [2.12; 50.56] *	2.48 [0.55; 14.38]	0.75 [0.23; 2.44]	0.42 [0.12; 1.45]
Asia	327	44.6	55.4	3.73 [0.96; 19.14]	3.55 [0.89; 18.99]	2.16 [0.89; 5.73]	1.51 [0.66; 3.68]
Europe	49	55.1	44.9	2.75 [0.58; 15.91]	2.31 [0.50; 13.24]	1.77 [0.50; 6.33]	0.68 [0.18; 2.48]
Maghreb	84	47.6	52.4	4.14 [0.84; 24.91]	3.38 [0.66; 21.36]	1.92 [0.68; 5.68]	0.89 [0.30; 2.69]
Immigration Status							
Residence permit	220	47.3	52.7	Ref.	Ref.	Ref.	Ref.
Undocumented	548	41.4	58.6	0.70 [0.39; 1.28]	0.92 [0.50; 1.71]	1.09 [0.67; 1.78]	0.41 [0.22; 0.75] *
Asylum Seeker	588	42.3	57.7	0.50 [0.27; 0.94] *	0.73 [0.39; 1.41]	0.59 [0.35; 1.01]	0.32 [0.17; 0.60] *
Length of Stay							
<3 months	218	45.4	54.6	Ref.	Ref.	Ref.	Ref.
3 months–5 years	791	39.2	60.8	2.10* [1.14; 4.01] *	0.70 [0.37; 1.34]	0.75 [0.47; 1.19]	0.88 [0.49; 1.63]
>5 years	346	49.1	50.9	2.25 [1.08; 4.85] *	0.72 [0.35; 1.49]	0.63 [0.35; 1.16]	0.63 [0.32; 1.26]
Housing conditions							
Unstable	715	37.6	62.4	Ref.	Ref.	Ref.	Ref.
Stable	641	48.5	51.5	1.78 [1.18; 2.70] *	1.96 [1.26; 3.08] *	1.29 [0.91; 1.82]	1.57 [1.04; 2.39] *
Income satisfaction							
Not Enough	1227	40.3	59.7	Ref.	Ref.	Ref.	Ref.
Enough	129	65.9	34.1	1.39 [0.76; 2.54]	1.22 [0.69; 2.17]	0.98 [0.46; 2.04]	0.47 [0.21; 0.99] *
Job							
No	874	45.8	54.2	Ref.	Ref.	Ref.	Ref.
Yes	482	37.3	62.7	1.32 [0.84; 2.06]	1.46 [0.89; 2.41]	1.82 [1.26; 2.62] *	2.16 [1.35; 3.49] *
Insurance coverage							
Yes	364	50.0	50.0	Ref.	Ref.	Ref.	Ref.
No	992	40.1	59.9	1.38 [0.59; 3.66]	1.80 [0.79; 4.52]	0.65 [0.29; 1.53]	1.67 [0.76; 3.96]

aOR: adjusted Odds Ratio, 95%CI: 95% Confidence Intervals, * *p* < 0.05.

**Table 3 ijerph-16-04878-t003:** Effect of length of stay (LOS) and Human Development Index; Effect modifier of length of stay on wealthy country of origin (COO) regarding physical perceived health in women.

No interaction		With Interaction; Effect Modifiers	
Probability of Being in Good Self-Perceived Health			
	aOR [95% CI]		aOR [95% CI]
Effect of wealthy COO vs. poor COO	1.94 * [1.06; 3.56]	Effect of wealthy COO vs. poor COO when LOS <3 months	5.96 * [1.65; 21.49]
Effect of stay longer than 3 months vs. stay shorter than 3 months	1.96 * [1.05; 3.65]	Effect of stay longer than 3 months vs. stay shorter than 3 months when women are from poorer countries	4.72 * [1.51; 14.72]
		Effect of stay longer than 3 months vs. stay shorter than 3 months when women are from wealthy countries	1.22 [0.59; 2.55]

The star (*) shows when 1 is outside the confidence interval meaning the aOR is significant. Interpretation: when women have stayed less than 3 months, those from wealthier countries are 5.96 times more likely to be in good physical health compared to women from poorer countries. OR are adjusted on all the characteristics in Table 3. aOR: adjusted Odds Ratio, 95% CI: 95% Confidence Intervals.

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
