# Peer review of "Self-Perceived Health among Migrants Seen in Médecins du Monde Free Clinics in Europe: Impact of Length of Stay and Wealth of Country of Origin on Migrants’ Health"

_ijerph, 2019, doi:10.3390/ijerph16244878_

Round 1

Reviewer 1 Report

I am satisfied with the efforts of the authors to address my comments and feedback.

Author Response

Thank you,

Reviewer 2 Report

The paper is solid and well-thought out. It certainly advances our understanding of the health of migrants in Europe.

However, I have a couple of suggestions:

Results - Table 1, line 172 and 173: I do think that it would be useful and clearer for the reader to report the single p-values in a separated column in the table instead of referring to them in a general/cumulative footnote. Results - lines 178, 185 and 188: the adjective "significant" should be supported by numerical/quantitative data (e.g. aOR, 95% CI, p-value). Results - Table 2, line 206: I do think that it would be useful and clearer for the reader to report the single p-values in a separated column in the table instead of referring to them in a general/cumulative footnote. Results - Table 3: please add footnotes for the acronyms aOR and 95% CI. Results - Table 3, lines 226-228: In my opinion the footnote is not easily readable. Please try to explain better. Results, lines 252-253: "Most of the literature concerns the health status of asylum seekers, refugees, and undocumented migrants". Which literature are you referring to? Could you explain better? Results, line 280: I think that "effects" should be replaced by "effect". Discussion, lines 289-297: some relevant and more recent papers about occupational health and safety of migrants are missing (see below), please consider to add them to the references and to briefly discuss them in the discussion section.

·   Moyce, S.C.; Schenker M. Migrant Workers and Their Occupational Health and Safety. Annu Rev Public Health 2018, 39, 351–365.

·         Sterud, T.; Tynes, T.; Sivesind Mehlum, I.; Veiersted, K.B.; Bergbom, B.; Airila, A.; Johansson, B.; Brendler-Lindqvist, M.; Hviid, K.; Flyvholm M.-A. A systematic review of working conditions and occupational health among immigrants in Europe and Canada. BMC Public Health 2018, 18, 770.

·         Hargreaves, S; Rustage K; Nellums, L.B.; McAlpine, A.; Pocock, N.; Devakumar, D.; Aldridge, R.W.; Abubakar, I.; Kristensen, K.L.; Himmels, J.W.; Friedland, J.S.; Zimmerman, C. Occupational health outcomes among international migrant workers: a systematic review and meta-analysis. Lancet Glob Health 2019, 7, e872-e882.

·         Arici C, Ronda-Pérez E, Tamhid T, Absekava K, Porru S. Occupational Health and Safety of Immigrant Workers in Italy and Spain: A Scoping Review. Int J Environ Res Public Health. 2019 Nov 11;16(22). pii: E4416. doi: 10.3390/ijerph16224416.

9. Discussion, “Limitations”: I think that the authors should also acknowledge the limitation that the study is based only on self-reported/perceived health data and not on clinical/objective ones.

10. Discussion, “Future Studies”: I warmly suggest to the authors to highlight that there is also a need for future studies/research based on objective/clinical data, in order to better characterize the health status of migrants.

Author Response

The paper is solid and well-thought out. It certainly advances our understanding of the health of migrants in Europe. However, I have a couple of suggestions:

Results - Table 1, line 172 and 173: I do think that it would be useful and clearer for the reader to report the single p-values in a separated column in the table instead of referring to them in a general/cumulative footnote.

We now report the pvalues.

Results - lines 178, 185 and 188: the adjective "significant" should be supported by numerical/quantitative data (e.g. aOR, 95% CI, p-value).

This has been changed.

Results - Table 2, line 206: I do think that it would be useful and clearer for the reader to report the single p-values in a separated column in the table instead of referring to them in a general/cumulative footnote.

We have to disagree on this as we report Confidence Intervals and reporting pvalues would be reporting exactly the same information. A scientific article should only report what is necessary and not lose a reader with twice the same information. As this paper is framed for a public health audience, Confidence Intervals are the most widely used framing for statistical significance we think this enough.

Results - Table 3: please add footnotes for the acronyms aOR and 95% CI.

Yes we added the explanation.

Results - Table 3, lines 226-228: In my opinion the footnote is not easily readable. Please try to explain better.

We have tried hard to make sure that this is readable already. We have changed the beginning of the footnote though we do not think there is clearer explanation for interaction results. If the reviewer has a suggestion we are more than ready to change again.

Results, lines 252-253: "Most of the literature concerns the health status of asylum seekers, refugees, and undocumented migrants". Which literature are you referring to? Could you explain better?

Yes indeed this was not very clear, we reformulated this into “The literature that concerns the health status of asylum seekers, refugees, and undocumented migrants shows that these three administrative categories face risk factors for mental health disorders during premigration (persecution, armed conflicts, and economic hardship), perimigration (different kinds of violence, life-threatening conditions, separation from family and support network) and post migration [for example see 17,31,32].”

Results, line 280: I think that "effects" should be replaced by "effect".

Indeed, thank you.

Discussion, lines 289-297: some relevant and more recent papers about occupational health and safety of migrants are missing (see below), please consider to add them to the references and to briefly discuss them in the discussion section. Moyce, S.C.; Schenker M. Migrant Workers and Their Occupational Health and Safety. Annu Rev Public Health 2018, 39, 351–365. Sterud, T.; Tynes, T.; Sivesind Mehlum, I.; Veiersted, K.B.; Bergbom, B.; Airila, A.; Johansson, B.; Brendler-Lindqvist, M.; Hviid, K.; Flyvholm M.-A. A systematic review of working conditions and occupational health among immigrants in Europe and Canada. BMC Public Health 2018, 18, 770. Hargreaves, S; Rustage K; Nellums, L.B.; McAlpine, A.; Pocock, N.; Devakumar, D.; Aldridge, R.W.; Abubakar, I.; Kristensen, K.L.; Himmels, J.W.; Friedland, J.S.; Zimmerman, C. Occupational health outcomes among international migrant workers: a systematic review and meta-analysis. Lancet Glob Health 2019, 7, e872-e882. Arici C, Ronda-Pérez E, Tamhid T, Absekava K, Porru S. Occupational Health and Safety of Immigrant Workers in Italy and Spain: A Scoping Review. Int J Environ Res Public Health. 2019 Nov 11;16(22). pii: E4416. doi: 10.3390/ijerph16224416.

Thank you for the references, we modified the discussion in order to introduce 2 of those articles (Arici et al., 2019; Moyce and Schenker, 2018) and we also introduce another article from Australian data showing some different effects as we thought it to be very interesting to show the disparity across countries (Reid et al., 2016).

Discussion, “Limitations”: I think that the authors should also acknowledge the limitation that the study is based only on self-reported/perceived health data and not on clinical/objective ones.

Indeed this should be acknowledged, there is however strong evidence regarding Self perceived health being a good measure of health status. In an earlier version of the article, we had written the paragraph below and following this remark, we decided to include it in the introduction and refer to it in the discussion.

“To assess health we use Self-Perceived Health (SPH). There is a strong body of evidence showing that SPH predicts mortality and morbidity as well as a medical diagnosis (DeSalvo et al., 2006; Doiron et al., 2014; Grant et al., 1995; Idler and Kasl, 1995; Krijger et al., 2014; McCallum et al., 1994; Yu et al., 1998). Across socio-economic status, SPH predicts less well mortality among higher socio economic statuses (Singh-Manoux et al., 2007) which might be due to richer and more educated reporting worse health when given a health state to assess (Bago d’Uva et al., 2008). SPH has been shown to be sensitive to ethnicity (Jürges, 2007) but Chandola and Jenkins show that there is no joint effect of SPH and ethnicity predicting morbidity (Chandola and Jenkinson, 2000), thus showing that the predictability of SPH is not modified by ethnicity. Moreover, SPH has been widely used to compare health in different countries (Doorslaer and Koolman, 2004; Mackenbach et al., 2008). A recent article shows that the effect of being a non-EU citizen in the EU on SPH is similar on chronic conditions or reporting limitations in daily life (Giannoni et al., 2016).”

Discussion, “Future Studies”: I warmly suggest to the authors to highlight that there is also a need for future studies/research based on objective/clinical data, in order to better characterize the health status of migrants.

We thank you for this suggestion, indeed this should be noted. We take this opportunity to highlight that ‘objective’ measures of health are still based on two subjective reasoning. First health professionals as any individual have their own subjectivity and pre conception about the patients that may even lead to discrimination in hopefully rare cases (Cognet et al., 2012) and that the diagnosis of clinical/objective measures is also based on patients perceptions as patients are not cars and they have their say in how much they suffer, where and how/when the condition they have deteriorated. Therefore what is called objective measure is actually based on two subjectives the one of the patient and the one of the professional (see Luchini, 2003 Encadré 1 page 17).

Arici, C., Ronda-Pérez, E., Tamhid, T., Absekava, K., Porru, S., 2019. Occupational health and safety of immigrant workers in Italy and Spain: a scoping review. International journal of environmental research and public health 16, e16224416. https://doi.org/10.3390/ijerph16224416

Bago d’Uva, T., Van Doorslaer, E., Lindeboom, M., O’Donnell, O., 2008. Does reporting heterogeneity bias the measurement of health disparities? Health Econ 17, 351–375. https://doi.org/10.1002/hec.1269

Chandola, T., Jenkinson, C., 2000. Validating Self-rated Health in Different Ethnic Groups. Ethnicity & Health 5, 151–159. https://doi.org/10.1080/713667451

Cognet, M., Hamel, C., Moisy, M., 2012. Santé des migrants en France : l’effet des discriminations liées à l’origine et au sexe. Revue européenne des migrations internationales. https://doi.org/10.4000/remi.5863

DeSalvo, K.B., Bloser, N., Reynolds, K., He, J., Muntner, P., 2006. Mortality Prediction with a Single General Self-Rated Health Question. J Gen Intern Med 21, 267–275. https://doi.org/10.1111/j.1525-1497.2005.00291.x

Doiron, D., Fiebig, D.G., Johar, M., Suziedelyte, A., 2014. Does self-assessed health measure health? Applied Economics 47, 180–194. https://doi.org/10.1080/00036846.2014.967382

Doorslaer, E.V., Koolman, X., 2004. Explaining the differences in income-related health inequalities across European countries. Health economics 13, 609–628.

Giannoni, M., Franzini, L., Masiero, G., 2016. Migrant integration policies and health inequalities in Europe. BMC Public Health 16, 1–14. https://doi.org/10.1186/s12889-016-3095-9

Grant, M.D., Piotrowski, Z.H., Chappell, R., 1995. Self-reported health and survival in the Longitudinal Study of Aging, 1984–1986. Journal of clinical epidemiology 48, 375–387.

Idler, E.L., Kasl, S.V., 1995. Self-Ratings of Health: Do they also Predict change in Functional Ability? J Gerontol B Psychol Sci Soc Sci 50B, S344–S353. https://doi.org/10.1093/geronb/50B.6.S344

Jürges, H., 2007. True health vs response styles: exploring cross-country differences in self-reported health. Health Economics 16, 163–178.

Krijger, K., Schoofs, J., Marchal, Y., Van de Vijver, E., Borgermans, L., Devroey, D., 2014. Association of objective health factors with self-reported health. J Prev Med Hyg 55, 101–107.

Luchini, S., 2003. De la singularité de la méthode d’évaluation contingente. Économie et statistique 357–358, 141–152.

Mackenbach, J.P., Stirbu, I., Roskam, A.-J.R., Schaap, M.M., Menvielle, G., Leinsalu, M., Kunst, A.E., 2008. Socioeconomic Inequalities in Health in 22 European Countries. New England Journal of Medicine 358, 2468–2481. https://doi.org/10.1056/NEJMsa0707519

McCallum, J., Shadbolt, B., Wang, D., 1994. Self-rated health and survival: a 7-year follow-up study of Australian elderly. Am J Public Health 84, 1100–1105. https://doi.org/10.2105/AJPH.84.7.1100

Moyce, S.C., Schenker, M., 2018. Migrant Workers and Their Occupational Health and Safety. Annu Rev Public Health 39, 351–365. https://doi.org/10.1146/annurev-publhealth-040617-013714

Reid, A., Peters, S., Felipe, N., Lenguerrand, E., Harding, S., 2016. The impact of migration on deaths and hospital admissions from work-related injuries in Australia. Australian and New Zealand Journal of Public Health 40, 49–54. https://doi.org/10.1111/1753-6405.12407

Singh-Manoux, A., Dugravot, A., Shipley, M.J., Ferrie, J.E., Martikainen, P., Goldberg, M., Zins, M., 2007. The association between self-rated health and mortality in different socioeconomic groups in the GAZEL cohort study. International Journal of Epidemiology 36, 1222–1228. https://doi.org/10.1093/ije/dym170

Yu, S.H., Kean, Y.M., Slymen, D.J., Liu, W.T., Zhang, M., Katzman, R., 1998. Self-perceived health and 5-year mortality risks among the elderly in Shanghai, China. American Journal of Epidemiology 147, 880–890.
